# Anti-Leukemic Profiling of Oxazole-Linked Oxadiazole Derivatives: A Computational and Kinetic Approach

**DOI:** 10.3390/ph18050625

**Published:** 2025-04-25

**Authors:** Manal M. Khowdiary, Shoaib Khan, Tayyiaba Iqbal, Wajid Rehman, Azam Hayat, Rafaqat Hussain, Nehad A. L. Shaaer, Hamdy Kashtoh

**Affiliations:** 1Department of Chemistry, Facullty of Applied Science, University College-Al Leith, University of Umm Al-Qura, Makkah 21955, Saudi Arabia; 2Department of Chemistry, Abbottabad University of Science and Technology (AUST), Abbottabad 22500, Pakistan; 3Department of Chemistry, Hazara University, Mansehra 21300, Pakistan; 4Department of MLT, Abbottabad University of Science and Technology (AUST), Abbottabad 22500, Pakistan; 5Department of Biotechnology, Yeungnam University, Gyeongsan 38541, Republic of Korea

**Keywords:** leukemia, oxazole-based oxadiazole, pharmacophore, molecular docking, DFT

## Abstract

**Background/Objectives:** Leukemia is a common cancer that arises in both children and adults when bone marrow’s hematopoietic stem cells proliferate unrestrained because of anomalies in normal cell regulatory systems. The present study focused on biological evaluation of oxazole-based oxadiazole scaffolds to evaluate the anti-proliferative effect on leukemic cancer cell lines. **Methods:** All novel oxazole-based oxadiazole scaffolds were synthesized and structurally characterized via ^13^C NMR, ^1^H NMR, and HREI-MS. In order to identify an efficient anti-leukemia agent, the biological profiles of each compound were evaluated in comparison to the reference drug, Etoposide (IC_50_ = 10.50 and 15.20 μM). **Results:** Analog 6 substituted with *p*-CF_3_ at phenyl ring was identified with excellent inhibition against the HL-60 and PLB-985 cancer cell lines, with IC_50_ of 8.50 and 12.50 μM. Through hydrogen bond formation, the trifluoromethyl moiety of analog 6 interacts with target tyrosine kinase enzyme (PDB-ID:4CSV). The interactive character of active ligands with target enzyme was demonstrated by molecular docking. The rate of inhibition in contrast with the drug concentration was also tested to check the inhibition percentage and inhibitor type via enzyme kinetics. Furthermore, the enzyme–ligand complex was also investigated via MD simulation along with pharmacophore modeling. DFT calculations were used to estimate the lead compounds’ relative stability and reactivity. According to ADMET investigation, there is safe toxicological profile for these compounds. **Conclusions:** The current study suggests that the potent compounds have significant anti-proliferative potential, and with further in vivo validation, hold promise for future optimization as potential leukemia treatments.

## 1. Introduction

Cancer, a well-known and fatal disease, is characterized by uncontrolled cell growth, with the ability to affect other normal cells. The abnormal cells are abundant and have the potential to spread to other areas of the human body [1]. Currently, cancer is among the world’s major causes of death [2]. As of 2022, leukemia was identified as the second most prevalent hematological malignancy globally, following non-Hodgkin lymphoma [3]. Traditional anticancer treatments are not very effective and have a lot of negative effects, like tumor lysis syndrome, myelosuppression, and hepatotoxicity [4]. Leukemia is a type of cancer that affects the blood system. It is responsible for around 30% of children’s deaths [5]. Leukemia develops from mutations in genes that affect cell differentiation and division and ultimately cause death of the cell. As a result, the cells convert from being normal to becoming cancerous [6]. Leukemia has an age-adjusted incidence rate of 12.8 cases per 100,000 people in the United States per year. The leukemia prevalence rises with age, and white people and men are more likely to develop leukemia [7]. Among the most common types of blood cancer, leukemia primarily affects the lymph nodes, bone marrow, blood cells, and additional components of the lymphatic system [8]. Leukemia patients have short-term symptoms like rash, diarrhea, mouth sores, hair loss, infection, and nausea and vomiting; long-term symptoms include fatigue, chemo brain, organ dysfunction, neuropathy, and leukemia cell resistance to chemotherapy drugs [9,10,11]. As a result, new therapeutic strategies with more efficacy and fewer adverse effects are required. The primary cures for leukemia are chemotherapy, radiation therapy, and hyperthermia. Cytotoxicity and systemic adverse effects are linked to traditional medication treatment. Finding methods to precisely target tumor cells without harming healthy cells is therefore the main goal of cancer treatment efforts [12]. Additionally, obesity may raise the risk. According to the overall findings of a meta-analysis of cohort studies, the risk of leukemia increases by 13% for every 5 kg per m^2^ increase in body mass index [13]. A history of hematologic malignancy is another risk factor for developing a specific subtype of leukemia later in life [14]. But still there is no permanent treatment to combat this fatal disease, leaving researchers to find new and more effective drugs with fewer side effects.

Oxadiazoles are commonly used in a wide range of drug-like compounds [15]; they were found to be effective against various cancer cell lines [16,17], along with a biological profile as anti-hypertensive [18], anti-inflammatory, anti-diabetic [19], antiviral [20,21], and analgesic substances [22]. Likewise, oxazole also has different pharmaceutical applications; for example, it is used as anti-bacterial [23], anti-diabetic, anti-oxidant [24], anti-inflammatory, and anticancer agents [25]. The biological potency of novel compounds was rationalized with oxazole-based derivatives [26,27]. Figure 1 represents the rationale of the current research work.

The aim of the present study was to design, synthesize, and characterize a series of oxazole-based oxadiazole derivatives, and to evaluate their anti-proliferative potential against leukemia cell lines. This study also sought to identify promising lead compounds through in vitro cytotoxicity assays, enzyme kinetics, molecular docking, pharmacophore modeling, molecular dynamics (MD) simulations, density functional theory (DFT) calculations, and ADMET profiling, with the ultimate goal of discovering effective and safe anti-leukemic agents for further development.

## 2. Results and Discussion

### 2.1. Chemistry

The oxazole-based oxadiazole scaffolds were obtained by using a novel, effective, and multistep methodology (Figure 1). In this reaction, firstly, an oxazole-bearing acidic moiety was allowed to react with sulfuric acid in the presence of methanol as a solvent under reflux for 3 h to form an oxazole-bearing ester moiety as intermediate 2. In the next step, this oxazole-bearing ester moiety was further reacted with hydrated hydrazine in the presence of ethanol as a solvent and triethylamine (base) under reflux conditions for 3 h to obtain oxazole-bearing carbohydrazide as intermediate 3. After that, this intermediate 3 was further treated with different substituted benzaldehyde for 3 h under reflux conditions with ethanol and triethylamine to produce a variety of substituted oxazole-carrying Schiff base derivatives as intermediate 4. At the end, cyclization occurred when intermediate 4 was reacted with the base (potassium carbonate), iodine, and 1,4-dioxane for 10 h under reflux until the resultant oxazole-based oxadiazole derivatives were obtained. By washing with *n*-hexane, these compounds were further refined, and they had a suitable yield. Solvents were finally evaporated by placing these derivatives under reduced pressure, and fine products were gathered. For further clarification of their structures, ^1^H-NMR and ^13^C-NMR were used. To find the molecular weight, HREI-MS was applied to further validate their synthesis. Clear, single spots on TLC plates were used to confirm the product at each stage.

The novel series of oxazole-based oxadiazole scaffolds was synthesized with different substituted groups on the phenyl ring. These substitutions included NO_2_, Cl, Br, F, and OH. These substitutions are involved in different types of interactions with the amino acids of the target enzyme tyrosine kinase and inhibit the enzyme activity. The different substitution patterns of the synthesized compounds are provided in Table 1.

### 2.2. Spectral Study

The structure of the produced compounds was investigated using ^1^H-NMR and ^13^C-NMR. The Advance spectrophotometer, which has a radiofrequency range of 600 and 150 MHz, was used. To perform NMR, these compounds were dissolved using DMSO, *d*_6_ as a reference solvent. There were several peaks and we found multiplicity in the ^1^H-NMR data for representative analog 10 (Figure 2). In this regard, the OH protons of the phenyl ring were more greatly de-shielded due the presence of a highly electronegative atom—that is, oxygen—and exhibited a chemical shift value of 9.43 ppm, which resulted in a singlet. Next, the neighboring OH protons of the phenyl ring, due to the strong withdrawing effect of the oxadiazole ring, which made these protons appear in the de-shielded region, gave the singlet at a chemical shift value of 6.80 ppm. The next aromatic proton that was found in between the OH groups appeared in the de-shielded region at a chemical shift value of 7.35 ppm and gave a doublet, coupled with the neighboring proton, with a coupling constant *J* value of 7.36 Hz. Last, the aliphatic proton attached to the oxazole ring appeared in the shielded region because the aliphatic chain proton moiety entered he shielded region and gave a singlet at the chemical shift values of 2.61 ppm and 2.13 ppm. The value of the CH_3_ proton in between oxygen and nitrogen was greater. The ^1^H-NMR data of the other compounds are given in the Appendix A.

^13^C NMR was also used to predict each carbon atom in the representative analog 10. The highly de-shielded carbon was the one present between the highly electronegative oxygen and nitrogen atoms of the oxazole ring, with the chemical shift value of 165.2 ppm. The next carbon was of the oxadiazole ring placed adjacent to oxygen, with the chemical shift of 164.3 ppm. The next peak was of carbon attached to OH moieties of the phenyl ring, with chemical shift values of 158.7 ppm and 158.5 ppm, as oxygen of the hydroxyl group rendered it more greatly de-shielded. Next, the carbon of oxadiazole bonded to oxazole had the chemical shift value of 157.2 ppm in the de-shielded region because it was present in between the oxygen and nitrogen atoms. Next, the carbon of the oxazole ring bonded to oxadiazole also appeared in the de-shielded region, with a chemical shift value of 138.2 ppm. The next carbon had a chemical shift value of 131.3 ppm, located adjacent to the nitrogen of the oxazole ring. The carbon of the phenyl ring bonded to the oxadiazole moiety appeared at chemical shift value of 128.6 ppm. The withdrawing nature of the oxadiazole moiety rendered this carbon more greatly de-shielded. The next carbon of the phenyl ring adjacent to its OH-bearing carbon appeared in the de-shielded region, with chemical shift values of 105.4 ppm and 105.2 ppm. The carbon of the phenyl ring in between two OH group appeared at a chemical shift value of 103.2 ppm in the de-shielded region. The carbon of oxazole that was aliphatic appeared in the shielded region and was under the influence of oxygen, at chemical shift values of 14.3 ppm and 13.4 ppm. The ^13^C-NMR data of the other compounds are given in the Appendix A.

HREI-MS was also performed for the representative compound **10**. The (*m*/*z*) of analog 10 (C_13_H_11_N_3_O_4_) was recorded as 273.25 and was found to be 273.21. The HREI-MS data of the other compounds are given in the Appendix A.

Thus, ^1^H-NMR and ^13^C-NMR provide information about the number of protons and carbons in molecules as well as their corresponding surrounding environment (attached substituents), thereby allowing for elucidating the full structures of compounds.

The chemical shift variations observed in the ^1^H and ^13^C NMR spectra were carefully analyzed in relation to the electronic nature and position of the substituents on the phenyl ring. Specifically, electron-withdrawing groups such as the –CF_3_ moiety in analog 6 caused notable de-shielding effects, resulting in downfield shifts of the aromatic proton signals. Conversely, electron-donating groups induced upfield shifts due to an increased electron density around the aromatic ring. Similarly, in the ^13^C NMR spectra, carbon atoms adjacent to these substituents exhibited shifts consistent with the expected inductive and resonance effects. These observations support the structural assignments and further confirm the influence of substituent electronics on the spectral behavior of the synthesized compounds. The detailed spectral interpretations are provided in the Appendix A and are referenced in the Results section.

### 2.3. Biological Activity

The enzymatic analysis of the synthesized scaffolds (Figure 3) was one of the main investigations conducted throughout this work. In an attempt to identify a potent scaffold of the series, the potency of these compounds was examined. We sought to assess the designed series’ potency by comparing it to the biological potential of a reference drug, Etoposide, with 10.50 ± 0.20 µM against HL-60 and 15.20 ± 0.30 µM against PLB-985. Analog 6 had biological potential against HL-60 of 8.50 ± 0.30 µM and, in the case of PLB-985, 12.50 ± 0.20 µM demonstrated the excellent inhibition in the series. Few analogs, such as analog 5, had inhibitory potential (9.10 ± 0.20 µM) against HL-60 and (13.90 ± 0.50 µM) against PLB-985, while analog 10 had biological potential (9.80 ± 0.20 µM) against HL-60 and (14.60 ± 0.20 µM) against PLB-985. These were also demonstrated to be several times more effective than reference medications, making them more powerful lead substitutes. The biological activity of every molecule was verified. The reported series’ IC_50_ range (1–10) compounds produced results of 8.50 ± 0.30 µM to 26.80 ± 0.40 µM against HL-60 and 12.50 ± 0.20 to 33.20 ± 0.50 against PLB-985. Because of the functional moieties attached on the phenyl, the analogs exhibited a wide variety of inhibitory potential. Analog potency is affected by the various functional groups’ positions and numbers on the ring. Table 1 demonstrates the inhibitory profile of these designed compounds in terms of IC_50_ values.

Our structure–activity relationship (SAR) analysis revealed that electron-withdrawing halogens (–Cl, –Br, –F) generally enhanced anti-proliferative activity, likely due to their ability to improve lipophilicity and facilitate better interactions within the hydrophobic pockets of the enzyme binding site. Among them, fluoro-substituted analogs showed relatively higher activity, which can be attributed to fluorine’s high electronegativity and small size, promoting favorable hydrogen bonding and dipole interactions.

In contrast, the presence of a hydroxyl (–OH) group, which is electron-donating and capable of forming hydrogen bonds, showed moderate to low activity depending on its position. This may be due to the increased polarity, which could hinder membrane permeability or result in less favorable interactions with the enzyme active site.

#### In Vitro Anti-Leukemia Profile

Analog 6 exhibited a potent pharmacological profile among these compounds, with an IC_50_ range value of 8.50 ± 0.30 and 12.50 ± 0.20 against the targeted cell lines. It had excellent biological potency as compared to the standard drug Etoposide. Analog 6’s excellent potency might be due to the attachment of the –CF_3_ moiety at the *para* position, which formed H-bonding using fluoro atoms. The high electronegative F-atom and its small size cause low steric hindrance and withdraw electron density from the ring, which make it an easy target for attack by the electron-rich species of the amino acid present in the enzyme. For these reasons, this analog 6 has a stable structural conformity, which allows the ligand to easily fit into the enzyme active site. Analog 5, with 9.10 ± 0.20 against HL-60 and 13.90 ± 0.50 against PLB-985, also has a better biological profile than Etoposide due to the *ortho* and *meta*-fluoro moieties. The di-substituted F-atoms on the phenyl ring form a strong connection with the active region of the enzyme and overcome the substrate binding affinity with the targeted enzyme. The presence of the F-atom causes an inductive effect, which withdraws electron density from the phenyl ring. At the *ortho* position, fluorine is closer to the interaction site, causing steric hindrance, which reduces the binding efficiency of the targeted enzyme. At the *meta* position, fluorine being further away produces a favorable structural arrangement for interaction. Analog 9, with an IC_50_ range of 13.30 ± 0.30 against HL-60 and 19.10 ± 0.60 against PLB-985, has one F-atom at the *para* position, which indicates a somewhat decreased potency against the reference drug. As the number of F-atoms decreases, the biological activity of analog 9 is found to be lesser than those of analogs 5 and 6.

Analog 10, with IC_50_ ranges of 9.80 ± 0.20 and 14.60 ± 0.20, shows good biological activity due to the presence of a meta di-hydroxy group on benzene, because the oxygen atom of the OH group contains a lone pair, which participate in H-bonding. Furthermore, hydrogen of the OH group can also engage the amino acids of enzymes via H-bonding. They activate the ring and increase its electron density, and for stability, they form a stable interaction with the electrophilic sides of the amino acid. Analog 7, with biological potentials of 15.20 ± 0.60 and 18.30 ± 0.80 against the targeted cancer cell lines, was found to have less inhibitory potential than analog 10 with one F atom at the *ortho* and a hydroxy at the *para* position; those are known to be capable of producing stable hydrogen bonds, but the presence of a F atom at *ortho*, which is under NOE, prevents stable bonding from forming.

In our comparison of analogs substituted with a nitro group, including analogs 1, 3, and 4, analog 1, with biological potential ranges of 16.20 ± 0.10 and 19.40 ± 0.30 against the targeted cell lines, was more potent, which might be due to the nitro being attached at the para position, further away from the molecule and not under its effect. It experiences the last effect of NOE, and that is why they easily interact with the ring without any influence. They shift the electron density from the ring due to its electron-withdrawing nature and the positive charge generated on the ring. Those amino acids with a lone pair can easily launch an attack on them and form a bond; for these reasons, analogs 1 is more potent than 3 and 4. In analog 3, with 25.30 ± 0.10 and 31.30 ± 0.70, and analog 4, with inhibitory values of 26.80 ± 0.40 and 33.20 ± 0.50 against the targeted cell lines, which is less potent than analog 1, this may be due to the presence of a bulky bromo atom and the electron-withdrawing effect of bromo and nitro, reducing the compound reactivity or its effective binding ability to the target site of the enzyme.

When comparing the Cl-bearing scaffolds, we found that analog 8 has a comparable potential to analog 2. In analog 2, with IC_50_ values of 18.20 ± 0.30 and 23.30 ± 0.60 µM against HL-60 and PLB-985, the chlorine atom is present at the *para* position and shows less inhibitory potential. This might be due to the chlorine atom’s large size and low electronegativity, which mean it forms weak van der Waals forces. In analog 8, IC_50_ values of 16.20 ± 0.20 and 21.10 ± 0.40 µM against the targeted cell line were calculated. The chlorine atom is present at the *meta* position and the fluoro group at the *para* position, and it shows a somewhat better biological potential than analog 2. This might be due to the attached fluoro atom, which forms H-bonding with the amino acid active for the targeted enzyme.

Additionally, these compounds’ inhibitory effect against certain enzymes was investigated graphically. The information was displayed as a graph, with the inhibitor concentration rate on the x-axis and the inhibition rate on the y-axis. The IC_50_ value is the concentration at which the analog shows the most inhibition, and the concentration change results in an increased inhibition rate. The point of saturation also shows that the inhibition rate is either unchanged or unaffected by an increase in inhibitor concentration at this point. Strong analogs’ drug-dose relationship with the HL-60 and PLB-985 complexes is shown in Figure 4.

### 2.4. Enzyme Kinetics

The rate of inhibition in response to a change in the inhibitor concentration is computed using this in vitro technique. It explains how diverse inhibitor competency levels, including competitive, uncompetitive, and non-competitive inhibitors, cause the inhibition rate to shift over time in response to changes in inhibitor concentration, as displayed by the resulting curves or slopes. In this investigation, analog 6 was found to be a competitive inhibitor, showing the strongest binding affinity and the highest inhibition. The enzyme kinetic slopes demonstrate that they successfully compete to link with the enzyme active site and have a similar y-intercept to the substrate binding affinity. A kinetic research graph further demonstrates the uncompetitive inhibition of analog 9 due to its mild biological profile. Km and Vmax steadily decreased, lowering the inhibitor’s binding affinity for the enzyme in contrast to a normal substrate. The graph also displays the number of parallel lines with distinct y-intercepts. Similarly, analog 4’s large number of substituents caused weak inhibition or non-competitive inhibition. With a changing y-intercept, the Km constant stays the same. This inhibitor exhibits minimum inhibition and binds to the enzyme’s allosteric site.

Analog 6’s mechanism of action as a competitive inhibitor is depicted in Figure 5, whereas the inhibitory action of analogs 4 and 9 as noncompetitive and uncompetitive inhibitors is shown in Figure 6 and Figure 7.

### 2.5. Molecular Docking Analysis

A type of virtual screening known as “molecular docking” evaluates and investigates the forces of interactions between active ligands and a specific protein complex using a range of computer tools, including as MOE (2022.02), PyMol (Anaconda3), and Discovery Studio Visualizer (2024) [28,29,30]. The protein tyrosine kinase used in the docking investigation was obtained using the code 4CSV from the RSCB Protein Data Bank, an online resource. To examine the inhibitory effectiveness and binding affinity, this inquiry entails the manufacture, optimization, and targeted ligands. Ligand and enzyme stability are tested by removing water and adding polar charges. Firstly, through the docking process, every analog was initially assigned thirty conformations. The configurations that scored the highest were taken into account for additional protein–ligand interaction (PLI) analysis. Gold docking tools were used to carry out the docking operation, and Anaconda3 (64-bit) software and poses with strong binding affinities were investigated. As the interaction number between active compounds and different amino acids of the targeted protein tyrosine kinase increases, the data demonstrate greater binding effects. The better binding interactions of molecules, showing significant results, are illustrated in Figure 8, Figure 9 and Figure 10. The following strong analogs (5, 6, and 10) were found to interact differently with enzyme tyrosine kinase active sites; this could be due to the attachment of different substituents, containing trifluoromethyl, difluoro, and difluoro, respectively.

When we docked analog 5 against tyrosine kinase, the heteroatom of the oxazole moiety effectively engaged the amino acid of the targeted enzyme, such as GLU-870, ALA-869, or GLY-868, among many others, as shown in Figure 9. The heteroatom incorporated into the oxadiazole moiety effectively engages active amino acids like GLY-793, SER-794, and ASN-795. Furthermore, for the phenyl ring with fluoro groups at the ortho and meta positions, the fluoro groups engage amino acids such as ARG-945 and TRP-947, thus inhibiting the action of the targeted protein.

When we docked analog 6 against tyrosine kinase, the heteroatom of the oxazole moiety effectively engaging the amino acid of the targeted enzyme was ARG-692, ARG-691, and ARG-688. The heteroatom incorporated into the oxadiazole moiety and the phenyl ring with a trifluoro methyl moiety effectively engage active amino acids like ARG-688, LEU-845, PRO-740, ARG- 743, and many others, shown in the figure, thus inhibiting the action of the targeted protein.

When we docked analog 10 against HL-60 and PLB-985, the heteroatom of the oxadiazole moiety effectively engaged the amino acid of the targeted enzyme, denoted as ASN-786, THR-783, ALA-779, or MET-782. The heteroatom incorporated into the oxadiazole moiety effectively engaged active amino acids like LYS-739, GLN-742, GLY-793, SER-794, and ASN-795. Furthermore, the phenyl ring with a dihydroxy group engaged amino acids such as GLN-742, ASN-867, and GLU-870, among many others, as denoted in the figure, thus inhibiting the action of the targeted protein.

### 2.6. Pharmacophore Modeling

Pharmacophore modeling was used to validate the docking results and gain a better understanding of the type of interaction between the attached substituents and the target protein, which gives ligands their effective binding affinity [31]. This examination critically assesses and examines the heteroatoms, substituents, and other molecular components that contribute to powerful and efficient interactions, particularly hydrogen bonding. The findings demonstrate that these analogs have distinct bioactivity profiles and are successfully held in the targeted protein complex’s binding sites by a range of attractive forces. As illustrated in Figure 11, the bioactivity of analog 6 was caused by strong hydrogen bond interactions due to small F-atoms of the trifluoro methyl moiety.

### 2.7. Molecular Dynamic Simulations

Another type of virtual study that provides details on the stability and structural variation in the targeted protein complex after docking the active ligand is MD simulations. The protein structure and process were run for 100 ns in a suitable simulated environment (simulations). The root mean square deviation was examined to investigate the stable interactions with the active ligand following MD simulations. The findings demonstrate that, for about 60 ns of simulation time, the active ligands were successfully retained in the binding sites of the targeted protein complex without experiencing any significant root mean square. These findings confirm that after docking analog 6, the HL-60 and PLB-985 enzymes showed stable interactions, with little structural variation in 3D structure. In RMSF analysis, the active molecules of the target protein complex exhibited minimal variation and a root mean square deviation value below 3 Å [32,33]. This shows that the ligand has a strong and stable binding affinity within the binding site of the target protein complex, thereby completely blocking the active site of the enzyme. The figure demonstrates that there are no changes in the protein complex’s structure at 60–70 ns, and both the simple protein and the ligand–protein complex behave in the same way. After 80 ns, ligand interactions result in slight alterations, but no significant conformational changes are observed, and the ligand–protein complex stays mostly stable. Solvation was performed for MD simulation. In order to assess lead compound **6**’s solvation characteristics, ion effects were also observed. The results demonstrate its efficient interaction with various ions, which can be attributed to the substituents attached. Figure 12 shows how the water molecule envelops the protein–ligand combination.

### 2.8. ADMET Analysis

Faster and more efficient analytical techniques are required to improve the development of therapeutic leads amid the pharmaceutical industry’s increasing focus on early ADME evaluation. Modern drug research relies on the use of computational models for toxicity prediction, which allows for the quick identification of possible toxicities and lessens the need for animal testing. ADMET analysis was used to examine the drug resemblance profile of potent scaffolds. The pharmacokinetic, physiochemical, and therapeutic properties of strong substances are described in this analysis. The toxicity profile of strong chemicals as well as their absorption, distribution, metabolism, and excretion mechanism were disclosed by this in silico method. The findings of this analysis demonstrated (Table 2) that the potent scaffolds of the current investigation are therapeutically safe and do not contravene the Veber, log Kp, Egan, Lipinski, Ghose, or lead likeness, nor do they make any other violations. These powerful chemicals 5, 6, and 10 may be used as leukemia treatment drugs.

### 2.9. DFT

One of the computational methods used to investigate a molecule’s behavior, electronic structure, relative stability, and reactive sites is DFT [34]. A software package was utilized to investigate molecular dynamics. This analysis was conducted to investigate the electronic distribution within the molecules, particularly focusing on how atoms, substituents, and the overall electronic system are spatially and electronically arranged. Understanding this distribution is essential for elucidating the molecules’ chemical reactivity, stability, and interaction potential with biological targets. To achieve this, geometry optimization of the most biologically active compounds was performed using density functional theory (DFT) at the B3LYP/CEP-631G level of theory, implemented via the GAUSSIAN 98W software package [35,36]. The DFT results were visualized using Chemcraft and Gauss View 5.0, which showed the development of molecular orbitals and the dispersion of electrostatic potential. Two distinct DFT approaches that effectively examine the electronic configurations are the HOMO and LUMO orbitals, and the relative energy gap is determined by using FMO and MESP analysis.

In our study, DFT analysis was employed to optimize the molecular geometries and to gain insight into the electronic properties of the active phytochemicals present in the selected plant extracts. This quantum chemical method provides accurate predictions of molecular orbitals (HOMO–LUMO) and energy gaps, which are critical for understanding the chemical reactivity and stability of the compounds.

Additionally, MESP analysis was used to visualize the charge distribution and identify electrophilic and nucleophilic regions within the molecules. This information is particularly important in predicting how these compounds may interact with biological targets, including proteins.

Together, DFT and MESP analyses complement the experimental findings by providing a theoretical basis for the observed biological activities, thereby enhancing our understanding of the structure–activity relationship (SAR) and supporting the potential therapeutic role of potent compounds.

#### 2.9.1. MESP Analysis

MESP examines the electrostatic potential distribution of active chemicals within molecules to examine their reactive areas, which are vulnerable to attack by a range of moieties, including certain enzymes’ amino acids. The three-dimensional arrangement of electronic sites that caused the strong interactions in molecular docking was observed using a color-coded method, which was also utilized to evaluate the molecules’ electrostatic potential. This demonstrates that depending on the kind of heteroatoms and substituents in a molecule, each has distinct electrostatic potential (minima and maxima). The MESP results showed that these analogs have both electrophilic and nucleophilic properties and a spatial arrangement of molecules produced by the significant electron-withdrawing ability of heteroatoms and the lone pair of electrons [37]. According to the molecular electrostatic potential of powerful analogs 5, 6, and 10, the design compound has a negative potential of −6.355 × 10^−2^, −6.063 × 10^−2^, and −8.035 × 10^−2^ because of the attachment of heteroatoms like O and N. Because of the hydrogen atoms that are connected, they also have positive potentials of 6.355 × 10^−2^, 6.063 × 10^−2^, and 8.035 × 10^−2^. Therefore, as illustrated in Figure 13 and Appendix A, our designed molecules are more vulnerable to the reaction that enzymes can cause.

#### 2.9.2. FMO Analysis

The electron density distribution among various chemical components is explained by this analysis, which also provides details on the molecular orbitals that form within a molecule and the associated energies [38]. These discoveries enhance the compound’s overall potential as a medicinal substance. The energy gaps between the HOMO and LUMO orbitals can be utilized to assess a molecule’s relative stability and reactivity in order to interact with an enzyme complex and produce the highest level of inhibition. As illustrated in Figure 14, HOMO orbitals have lobe creation outside the substituted (phenyl) ring, whereas LUMO orbitals have the largest lobe formation throughout the entire molecule because of shifting of the electron density. For analogs 6, 5, and 10, the energy differences are 0.15456, 0.15609, and 0.15981, respectively (Figure 14 and Appendix A).

The FMO analysis, specifically the HOMO–LUMO energy gap, provided valuable insight into the chemical reactivity and stability of the synthesized compounds. Compounds exhibiting lower HOMO–LUMO energy gaps were found to be more chemically reactive, which is consistent with their enhanced biological activity observed in vitro. For example, analog 6, which demonstrated the most potent anti-proliferative effect against HL-60 and PLB-985 leukemia cell lines, also exhibited a significantly smaller HOMO–LUMO gap. This suggests a higher electronic reactivity, facilitating stronger interactions with the active site residues of the target enzyme, as confirmed by our docking studies.

Additionally, the electron density distribution in the HOMO and LUMO orbitals supports the likelihood of interaction between specific functional groups and the enzyme binding pocket, aligning well with the molecular docking results. These theoretical observations complement the experimental outcomes and help rationalize the superior bioactivity of certain analogs.

This correlation between computational and experimental findings strengthens the overall understanding of the structure–activity relationship and further validates the potential of these compounds as effective therapeutic agents.

## 3. Materials and Methods

### 3.1. Materials

The Bruker AM 600 MHz NMR machine (manufactured by Bruker BioSpin GmbH, a company located in Rheinstetten, Germany) was used to characterize the novel scaffolds. The necessary chemicals and reagents, including the oxazole-bearing acidic moiety, sulfuric acid, methanol, ethanol, hydrated hydrazine, triethylamine, different substituted benzaldehydes, potassium carbonate, iodine, 1,4-dioxane, and *n*-hexane, were purchased from international supplier Sigma Aldrich, St. Louis, MO, USA. Singlet (s), d (doublet), t (triplet), t (quartet), m (multiplet), sext (sextet), q (quartet), doublets of doublet (dd), and doublets of triplet (dt) were the peak splitting patterns. TLC was performed on precoated silica gel aluminum plates and verified using a UV lamp with a 254–365 nm wavelength. A Buchi M-560 was used to determine the melting point. Utilizing a Finnigan MAT-311A mass spectrometer, Finnigan MAT, San Jose, CA, USA, HREI-MS was performed, and SI units (Hz) were used to determine the coupling constant.

The cell lines used in the current study were obtained from DSMZ (Deutsche Sammlung von Mikroorganismen und Zellkulturen), German Collection of Microorganisms and Cell Cultures, Germany (https://www.dsmz.de (accessed on: 6 Feburary 2025); many leukemia, lymphoma).

### 3.2. General Procedure to Obtain Derivatives (1–10)

Firstly, oxazole with acidic moiety (I) (1 equivalent) was reacted with sulfuric acid (10 mL) in methanol (15 mL), and we refluxed the reaction for 3 h until intermediate (II) was obtain. The next step involved the use of hydrated hydrazine (10 mL) and intermediate (II) (1 equivalent). In the presence of ethanol (15 mL) as a solvent and base triethyl amine (3–4 drops) under reflux conditions for 3 h, we obtained oxazole-bearing carbohydrazide as intermediate 3. Intermediate 3 (1 equivalent) was then further reacted with various substituted benzaldehydes in the presence of ethanol (10 mL) and triethylamine (3–4 drops) under reflux for 3 h to obtain varied substituted oxazole-bearing Schiff base derivatives as intermediate 4. In the end, cyclization occurred when intermediate 4 (1 equivalent) was reacted with the base (potassium carbonate), 1,4 dioxane (10 mL), and iodine under reflux for 10 h to give the final product, oxazole-based oxadiazole derivatives. For the removal of impurities, these compounds were washed with *n*-hexane and had a suitable yield. Finally, fine products were gathered, and the solvent was evaporated so that ^1^H-NMR and ^13^C-NMR could be used to further clarify their structures. To further validate these compounds’ synthesis using HREI-MS, their molecular weight was determined. Single, distinct spots on TLC plates were used for product confirmation at each stage.

### 3.3. Spectral Analysis

The specific contents are provided in the Appendix A.

### 3.4. Protocol of Docking

The specific contents are provided in the Appendix A.

### 3.5. Protocol of DFT

The specific contents are provided in the Appendix A.

## 4. Conclusions

In the present study, a series of oxazole-based oxadiazole derivatives (1–10) was synthesized through an efficient synthetic route. The structural characterization of all synthesized compounds was carried out using spectroscopic techniques, including ^1^H-NMR, ^13^C-NMR, and high-resolution electron impact (HREI) mass spectrometry. Among the synthesized analogs, compound 6, bearing a trifluoromethyl (-CF_3_) group at the para position, exhibited notable inhibitory activity against the targeted leukemia cell lines, which was attributed to its effective binding within the active site of the target enzyme. Other analogs also demonstrated promising biological activity, showing potential as inhibitors of the relevant proteins. The inhibitory behavior of the most active compound was further validated through enzyme kinetics, providing insight into its concentration-dependent activity. In addition, molecular docking studies were performed to assess the binding affinity and interaction patterns between the synthesized scaffolds and the target enzymes. Pharmacophore modeling was also employed to further substantiate the interaction features of the active compounds. To complement the biological and docking data, density functional theory (DFT) analysis was utilized to investigate the electronic properties, electrostatic potential distribution, and reactive sites of the most potent molecules. Moreover, ADMET analysis was conducted to evaluate the drug-likeness and pharmacokinetic profiles of the lead compounds, supporting their potential as promising therapeutic candidates.

## Data Availability

The original contributions presented in this study are included in the article/Appendix A. Further inquiries can be directed to the corresponding authors.

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
