# Peer review of "Anti-Leukemic Profiling of Oxazole-Linked Oxadiazole Derivatives: A Computational and Kinetic Approach"

_pharmaceuticals, 2025, doi:10.3390/ph18050625_

Round 1
Reviewer 1 Report
Comments and Suggestions for Authors
The work titled " Medicinal Potential of Oxazole-Connected Oxadiazole Derivatives: Computational Modeling, Kinetic Assessment and Anti-Leukemic Evaluation in HL-60 and PLB-985 cells" represents a significant advancement in the field of cancer treatment
In this study, the authors synthesized oxazole-based oxadiazole derivatives and evaluate their antiproliferative effects leukemic cancer cells, all of the synthesized compounds were well characterized by using 13C NMR, 1H NMR, and HRMS. Among the synthesized and evaluated compounds, compounds No6 showed significant activities with micromolar IC50 values. Additionally molecular docking studies were performed to confirm the binding interactions as well as molecular dynamic simulation, DFT calculations and ADMET investigations were conducted which all make this work valuable. All of these findings seem valuable and suitable contribution to be published in the pharmaceuticals journal after justifying some points:
- The title should be improved since it is not common to use phrase “medicinal Potential” because this phrase used usually for clinical candidates of compounds while this work has just in in vitro and virtual investigations
- Line 19 phrase “to trat leukemia ” is not suitable since we are in early stage of compound evaluation, it is recommended to use “to evaluate antiproliferative effect on leukemic cancer cell lines” as well as edit “structurally investigated using 13C NMR, 1H NMR, and HREI-MS” to “characterize “ line 22 you can use compound 6 instead of analog
- It recommended to remove the SD values from the IC50, as well as you should add the IC50 value for the reference drug
- Line 25 you mentioned that compound 6 was interacted with the active site of enzymes, which enzyme ??
- Is the using of ADMET just for the safety of these drug, may the safety could be considered by using a normal cell line in comparison with the cancer one by selectivity index while the ADMET should give you general predicted pharmacokinetics data
- In the end of the abstract amy could add conclusion for your study with some recommendation for your future work plan on this scaffold
- In the introduction a recent statistics form last WHO data for the cancer and leukemia could be added rather than using older one
- Figure 1 should be removed since this data was mentioned in the main text and no need to repeat that, while in figure 2 it is not common to references to the main figure the reference should be cited in the main text accordingly
- In the last paragraph of the introduction, you should add the rational behind this study the aims and what did you planned to do without references
- Regarding scheme 1 unify all bonds in the structures why there are some like bold and the other are not ?
- The discerption of NMR should be merged together and mention the HRMS results too briefly
- In Table 1 since you added the phenyl ring you just write the R groups only and instead of title compound write R group, as well as to this table add the footnote of the P vales for SEM or SD
- Why the authors were used Etoposide? there are any structure similarity? could you add the structure of this drug too ?
- All in vitro words should be written in italic style in the whole manuscript
- Line 170-171 you should be awarded regarding the difference between the inhibitory range and the IC50 range, this should be edited in the results
- In the results you still mention the targeted enzyme and amino acids without writing which enzyme is that? Or amino acids we are talking about ?
- Figure 5 have just IC50 values with the structures so we could not title it with SAR, and the data in this figure dose not different than table 1 so i recommended to remove it to improve it regrading the SAR data which mentioned in the main text
- In figure 6 you should improve the resolution and you can merge these two cell lines in the same figure and add the error bars accordingly, as well as add the concentrations unit to this figure
- Cell Viability section should be removed because it is the same results of cell inhibition oppositely so it doesn’t make sense
- There are misunderstanding and confusing results of molecular docking, you still mention “in the active site of the targeted enzyme” and I am still looking for this enzyme ?? you mention that you get the code form PDB but you did not mention the code? How could you write “Potent scaffold 5 molecular docking profile against HL-60 and PLB-985” do we making a docking for the whole cancer cell line ???? or we make the docking regarding specific target in this cell lines?? Like an enzyme or receptor ?? this very important point should be solved, otherwise all docking data are not valuable
- The predicted results of ADMET could be written as table better than figures
- The reasons behind using DFT and MESP analysis in this work should be will descried
- To the martial and method section you should write all used chemicals and reagents
- There is no supplementary file??
- The data of each synthesized compound should be written, including the IUPAC name, yields, HRMS, and NMR data !!
- The conclusion should not similar like the abstract, no need to write any values there, you should improve the conclusion well
Best wishes
Author Response
Comments and Suggestions for Authors
The work titled " Medicinal Potential of Oxazole-Connected Oxadiazole Derivatives: Computational Modeling, Kinetic Assessment and Anti-Leukemic Evaluation in HL-60 and PLB-985 cells" represents a significant advancement in the field of cancer treatment
In this study, the authors synthesized oxazole-based oxadiazole derivatives and evaluate their antiproliferative effects leukemic cancer cells, all of the synthesized compounds were well characterized by using 13C NMR, 1H NMR, and HRMS. Among the synthesized and evaluated compounds, compounds No6 showed significant activities with micromolar IC50 values. Additionally molecular docking studies were performed to confirm the binding interactions as well as molecular dynamic simulation, DFT calculations and ADMET investigations were conducted which all make this work valuable. All of these findings seem valuable and suitable contribution to be published in the pharmaceuticals journal after justifying some points:
- The title should be improved since it is not common to use phrase “medicinal Potential” because this phrase used usually for clinical candidates of compounds while this work has just in in vitro and virtual investigations
Reply: Title was improved as per kind suggestion.
- Line 19 phrase “to trat leukemia ” is not suitable since we are in early stage of compound evaluation, it is recommended to use “to evaluate antiproliferative effect on leukemic cancer cell lines” as well as edit “structurally investigated using 13C NMR, 1H NMR, and HREI-MS” to “characterize “ line 22 you can use compound 6 instead of analog
Reply: All the suggested corrections were made as per kind suggestion.
- It recommended to remove the SD values from the IC50, as well as you should add the IC50 value for the reference drug
Reply: Reply: All the suggested corrections were made as per kind suggestion.
- Line 25 you mentioned that compound 6 was interacted with the active site of enzymes, which enzyme ??
Reply: Potent compounds were interacted with the active site of tyrosine kinase enzymes to gain an insight into the binding interactions. These interactions were studied to validate biological activity profile of the potent compounds with antiproliferative effect. The target enzyme was mentioned in the manuscript as per kind suggestion.
- Is the using of ADMET just for the safety of these drug, may the safety could be considered by using a normal cell line in comparison with the cancer one by selectivity index while the ADMET should give you general predicted pharmacokinetics data
Reply: The ADMET predictions for the evaluated compounds suggest that they exhibit highly favorable pharmacokinetic characteristics. These include efficient absorption and distribution, notable metabolic stability, effective routes of excretion, and minimal toxicity. Collectively, these properties highlight the drug-likeness of the compounds and underscore their suitability for advancement in the drug development pipeline. The corresponding data supporting these findings are presented in Figure 16-18. These predictive insights are instrumental in narrowing down the most promising candidates for clinical evaluation and serve as a foundational step toward the rational design and development of effective therapeutic agents.
- In the end of the abstract amy could add conclusion for your study with some recommendation for your future work plan on this scaffold
Reply: Future work plan was added in abstract as per kind suggestion.
- In the introduction a recent statistics form last WHO data for the cancer and leukemia could be added rather than using older one
- Reply: recent statistics were added as per kind suggestion.
- Figure 1 should be removed since this data was mentioned in the main text and no need to repeat that, while in figure 2 it is not common to references to the main figure the reference should be cited in the main text accordingly
Reply: Figure 1 was removed and reference in figure 2 was cited in the main text as per kind suggestion.
- In the last paragraph of the introduction, you should add the rational behind this study the aims and what did you planned to do without references
Reply: Added as per kind suggestion.
- Regarding scheme 1 unify all bonds in the structures why there are some like bold and the other are not ?
Reply: Scheme 1 was made uniform as per kind suggestion.
- The discerption of NMR should be merged together and mention the HRMS results too briefly
Reply: Description of NMR was merged together and the HRMS results were also mentioned as per kind suggestion.
- In Table 1 since you added the phenyl ring you just write the R groups only and instead of title compound write R group, as well as to this table add the footnote of the P vales for SEM or SD
Reply: All the suggested changes were made in Table 1 as per kind suggestion. Moreover, the footnote of the P values for SEM was also added.
- Why the authors were used Etoposide? there are any structure similarity? could you add the structure of this drug too ?
Reply: Thank you for your insightful comment. Etoposide was selected as a standard reference compound in our study owing to its well-documented anticancer properties, as well as its extensive use in the literature as a positive control in assays evaluating anticancer agents. The inclusion of Etoposide provided a reliable benchmark, enabling us to compare the bioactivity of our synthesized compounds against a clinically relevant and pharmacologically active agent.
Although there is no direct structural similarity between Etoposide and synthesized scaffolds, its use was not based on structural analogy, but rather on its functional role as a reference standard. This comparative framework allowed us to better understand the therapeutic potential and relative efficacy of the novel compounds, thereby placing our findings within a broader pharmacological and biomedical context.
Moreover, the structure of drug Etoposide was also added in Figure 3 as per kind suggestion. Structural comparison of standard drug and novel compounds shows that the standard drug is large in size with high molecular weight. This may offer steric hindrance in biological reactivity against the target enzyme as anti-cancer agent. The novel compounds have different substitutions and is also small in size, which elevates the biological profile of these compounds as anti-cancer agents in comparison to standard drug.
- All in vitro words should be written in italic style in the whole manuscript
Reply: Corrected as per kind suggestion.
- Line 170-171 you should be awarded regarding the difference between the inhibitory range and the IC50 range, this should be edited in the results
Reply: these values represent the IC50 value, which was corrected in manuscript as per kind suggestion.
- In the results you still mention the targeted enzyme and amino acids without writing which enzyme is that? Or amino acids we are talking about ?
Reply: The interactions between potent compounds and different amino acids of the targeted protein tyrosine kinase were studied under molecular docking study, and the name of target protein tyrosine kinase was also included in manuscript as per kind suggestion. The different amino acids present on active site of tyrosine kinase enzyme were mentioned which are illustrated in 2D visualization of docking study (Figures 8-10).
- Figure 5 have just IC50 values with the structures so we could not title it with SAR, and the data in this figure dose not different than table 1 so i recommended to remove it to improve it regrading the SAR data which mentioned in the main text
Reply: Figure 5 was removed as per kind suggestion.
- In figure 6 you should improve the resolution and you can merge these two cell lines in the same figure and add the error bars accordingly, as well as add the concentrations unit to this figure
Reply: Figure 6 was revised as per kind suggestion.
- Cell Viability section should be removed because it is the same results of cell inhibition oppositely so it doesn’t make sense
Reply: this section was moved to supplementary information as per kind suggestion.
- There are misunderstanding and confusing results of molecular docking, you still mention “in the active site of the targeted enzyme” and I am still looking for this enzyme ?? you mention that you get the code form PDB but you did not mention the code? How could you write “Potent scaffold 5 molecular docking profile against HL-60 and PLB-985” do we making a docking for the whole cancer cell line ???? or we make the docking regarding specific target in this cell lines?? Like an enzyme or receptor ?? this very important point should be solved, otherwise all docking data are not valuable
Reply: The protein tyrosine kinase used in the docking investigation was obtained using the code 4CSV from the RSCB Protein Data Bank, an online resource. Moreover, all the necessary corrections were made in molecular docking section as per mind suggestion.
- The predicted results of ADMET could be written as table better than figures
Reply: ADMET results were presented in table as per kind suggestion.
- The reasons behind using DFT and MESP analysis in this work should be will descried
Reply: The reasons behind using DFT and MESP analysis in this work were described as per kind suggestion.
- To the martial and method section you should write all used chemicals and reagents
Reply: All the used chemicals and reagents were added in the material and method section as per kind suggestion..
- There is no supplementary file??
Reply: supplementary file was provided as per kind suggestion.
- The data of each synthesized compound should be written, including the IUPAC name, yields, HRMS, and NMR data !!
Reply: Data of each synthesized compounds including the IUPAC name, yields, HRMS, and NMR data was provided in supplementary file as per kind suggestion
- The conclusion should not similar like the abstract, no need to write any values there, you should improve the conclusion well
Reply: Conclusion was revised as per kind suggestion.
Best wishes
We highly appreciate and are grateful to you for your efforts in reviewing this manuscript and reshaping it to the best.
Kind Regards

Reviewer 2 Report
Comments and Suggestions for Authors
The research article under the title "Medicinal Potential of Oxazole-Connected Oxadiazole Deriva- 2
tives: Computational Modeling, Kinetic Assessment and Anti- 3
Leukemic Evaluation in HL-60 and PLB-985 cells" written by Khowdiary and coworkers presents results on the structural analysis and biological activity of newly obtained compounds. The manuscript shows novel results, with appropriate experimental and theoretical methods applied for the discussion of results. This manuscript is interesting to the readers of Pharmaceuticals, although the authors should answer the following questions. My recommendation is a Major revision.
The authors should consider the following:
- Analog 6 in the abstract should be explained with structural information; it is beneficial to have explicitly stated structural parameters that are important for the activity
- Line 42 - add and between 9.4 and 18%
- The authors should add the aim of the study in the last paragraph of the introduction
- Change newly synthetic potent compound to newly synthesized compound with potential activity
- pagraph in section 2.1 should also contain the description of the substituents on the last structure, as an introduction
- The authors should give an explanation of the NMR spectra with respect to the present substituents, it would be interesting to see if some of the positions were shifted due to the presence of various substituents
- for the biological activity, the authors should outline if some of the substituents affect the biological activity. It is very important to increase the number of references by comparing the obtained results with literature values
- in the molecular docking section the authors should clearly name the type of interactions formed between different groups within structure of analogs and amino acids
- Figures 16-18 should be moved to the supplementary information, as they are not very informative; the authors should explain these results in the main text
- the authors should mention the basis set used for the optimization and verify that the chosen level of theory is applicable to this group of compounds by citing the appropriate literature
- the results obtained in FMO analysis should be correlated with the experimental findings
- the supplementary information is not included in the submission, there is no need to present the theoretical methods in the supplementary information. The authors should move these to the main text, as it is much easier to follow the procedure.
Author Response
Comments and Suggestions for Authors
The research article under the title "Medicinal Potential of Oxazole-Connected Oxadiazole Deriva- 2
tives: Computational Modeling, Kinetic Assessment and Anti- 3
Leukemic Evaluation in HL-60 and PLB-985 cells" written by Khowdiary and coworkers presents results on the structural analysis and biological activity of newly obtained compounds. The manuscript shows novel results, with appropriate experimental and theoretical methods applied for the discussion of results. This manuscript is interesting to the readers of Pharmaceuticals, although the authors should answer the following questions. My recommendation is a Major revision.
The authors should consider the following:
- Analog 6 in the abstract should be explained with structural information; it is beneficial to have explicitly stated structural parameters that are important for the activity
Reply: Structural information of analog 6 was added in abstract as per kind suggestion.
- Line 42 - add and between 9.4 and 18%
Reply: added as per kind suggestion.
- The authors should add the aim of the study in the last paragraph of the introduction
Reply: Aim of the study was added in introduction as per kind suggestion.
- Change newly synthetic potent compound to newly synthesized compound with potential activity
Reply: corrected as per kind suggestion.
- pagraph in section 2.1 should also contain the description of the substituents on the last structure, as an introduction
Reply: Description of the substituents was provided in section 2.1 as per kind suggestion.
- The authors should give an explanation of the NMR spectra with respect to the present substituents, it would be interesting to see if some of the positions were shifted due to the presence of various substituents
Reply: Provided as per kind suggestion.
- for the biological activity, the authors should outline if some of the substituents affect the biological activity. It is very important to increase the number of references by comparing the obtained results with literature values
Reply: Added as per kind suggestion.
- in the molecular docking section the authors should clearly name the type of interactions formed between different groups within structure of analogs and amino acids
Reply: Type of interactions formed between different groups within structure of analogs and amino acid of target enzyme tyrosine kinase were added in docking section in figures 8 to 10 as per kind suggestion.
- Figures 16-18 should be moved to the supplementary information, as they are not very informative; the authors should explain these results in the main text
Reply: the data of figure 16 to 18 was provided in Table 2 as per kind suggestion.
- the authors should mention the basis set used for the optimization and verify that the chosen level of theory is applicable to this group of compounds by citing the appropriate literature
Reply: The basis set used for the optimization and the chosen level of theory was mentioned by citing the appropriate literature as per kind suggestion.
- the results obtained in FMO analysis should be correlated with the experimental findings
Reply: The results obtained in FMO analysis were correlated with the experimental findings as per kind suggestion.
- The supplementary information is not included in the submission, there is no need to present the theoretical methods in the supplementary information. The authors should move these to the main text, as it is much easier to follow the procedure.
Reply: supplementary file was provided in the submission as per kind suggestion.

Round 2
Reviewer 1 Report
Comments and Suggestions for Authors
The authors were improved their work well but still some points should be clarified:
- To the abstract you should edit the “tyrosine kinase enzyme” with the code which was used from PDB like (ID: 4CSV) since the tyrosine kinase enzyme is a large family of many enzymes
- In figure 2 remove the cited references since you mention these references in the main text, as well as unify all bonds in these structures, why some bonds are in bolds and the others are not, you could do that like what did you do in the scheme 1
- I think you should submit the work in the Journal style
- The n-hexane should be written like n-hexane, n should be in italic style
- Some typing errors should be edited like 3hours, 6.80ppm, 7.36Hz add spaces accordingly, Unify 13C-NMR in the whole manuscript
- Correct the inhibitory range to the IC50 range, this should be edited in the results
- Figure 14 it is not clear which one HOMO and LUMO for compound 6 or 5 or 10, as well as it is recommended to move the HOMO and LUMO of compounds 5 and 10 to the supp file
- Regarding the compound data which provided to the Supp. File it is recommended to write at least one or two compounds’ data in the main text and mention that the other data are available in the supp file
Best wishes
Author Response
Comments and Suggestions for Authors
The authors were improved their work well but still some points should be clarified:
- To the abstract you should edit the “tyrosine kinase enzyme” with the code which was used from PDB like (ID: 4CSV) since the tyrosine kinase enzyme is a large family of many enzymes
Reply: PDB Id of tyrosine kinase enzyme was mentioned in the abstract as per kind suggestion.
- In figure 2 remove the cited references since you mention these references in the main text, as well as unify all bonds in these structures, why some bonds are in bolds and the others are not, you could do that like what did you do in the scheme 1
Reply: Cited references were removed from figure 2. Moreover, all the bonds were unified as per kind suggestion.
- I think you should submit the work in the Journal style
Reply: The whole manuscript was made according to journal style as per kind suggestion.
- The n-hexane should be written like n-hexane, n should be in italic style
Reply: Corrected as per kind suggestion.
- Some typing errors should be edited like 3hours, 6.80ppm, 7.36Hz add spaces accordingly, Unify 13C-NMR in the whole manuscript
Reply: All the typing errors were removed. Moreover, 13C-NMR was also unified as per kind suggestion.
- Correct the inhibitory range to the IC50 range, this should be edited in the results
Reply: Corrected as per kind suggestion.
- Figure 14 it is not clear which one HOMO and LUMO for compound 6 or 5 or 10, as well as it is recom:mended to move the HOMO and LUMO of compounds 5 and 10 to the supp fil
Reply: Figure 14 was made clear by mentioning the number of compounds. Moreover, HOMO and LUMO of compounds 5 and 10 were moved to the supplementary file as per kind suggestion.
- Regarding the compound data which provided to the Supp. File it is recommended to write at least one or two compounds’ data in the main text and mention that the other data are available in the supp file
Reply: Compound data for representative compound 10 was given in main manuscript and it was also mentioned that the other data are available in the supplementary file as per kind suggestion.
Best wishes
Thank you for your feedback and for taking the time to review our manuscript and reshaping it to the best.
Kind Regards

Reviewer 2 Report
Comments and Suggestions for Authors
The authors have answered all of the questions properly. The manuscript is suitable for publication in the present form.
Author Response
The authors have answered all of the questions properly. The manuscript is suitable for publication in the present form.
Reply: Thank you for your positive feedback and for taking the time to review our manuscript and reshaping it to the best. We are pleased to hear that the revisions meet your expectations and that the manuscript is now considered suitable for publication.
Kind Regards
